# Interaction of the causal agent of apricot bud gall *Acalitus phloeocoptes* (Nalepa) with apricot: Implications in infested tissues

Shijuan Li[1], Muhammad Khurshid[2], Junsheng Yao[3], Jin Zhang[4], Mohammed Mujitaba Dawuda[5,6], Zeshan Hassan[7], Shahbaz Ahmad[8], Bingliang Xu[1]*

1 College of Plant Protection, Gansu Agricultural University, Lanzhou, China, 2 School of Biochemistry and Biotechnology, University of the Punjab, Lahore, Pakistan, 3 Gansu Agriculture Vocational and Technical College, Horticulture Technology, Lanzhou, Gansu, China, 4 College of Pastoral Agriculture Science and Technology, Lanzhou University, Lanzhou, Gansu, China, 5 College of Horticulture, Gansu Agricultural University, Lanzhou, Gansu, China, 6 Department of Horticulture, FoA, University for Development Studies, Tamale, Ghana, 7 College of Agriculture, Bahauddin Zakariya University, Multan, Bahadur Sub Campus Layyah, Pakistan, 8 Institute of Agricultural Sciences, University of the Punjab, Lahore, Pakistan

* xubl@gsau.edu.cn, 1153657458@qq.com

**Data Availability Statement:** All relevant data are within the paper and its Supporting Information files.

## Abstract

Apricot bud gall mite, *Acalitus phloeocoptes* (Nalepa), is a destructive arthropod pest that causes significant economic losses to apricot trees worldwide. The current study explores the ways to understand the mode of dispersal of *A. phloeocoptes*, the development and ultrastructure of apricot bud gall, and the role of phytohormones in the formation of the apricot bud galls. The results demonstrated that the starch granules in the bud axon were extended at the onset of the attack. During the later stages of the attack, the cytoplasm was found to deteriorate in infected tissues. Furthermore, we have observed that the accumulation of large amounts of cytokinin (zeatin, ZT) and auxin (indoleacetic acid, IAA) led to rapid bud proliferation during rapid growth period, while abscisic acid (ABA) controls the development of gall buds and plays a vital role in gall bud maturity. The reduction of gibberellic acid (GA3) content led to rapid lignification at the later phase of bud development. Overall, our results have revealed that the mechanism underlying the interaction of apricot bud gall with its parasite and have provided reliable information for designing valuable Apricot breeding programs. This study will be quite useful for pest management and will provide a comprehensive evaluation of ecology-based cost-effective control, life history and demographic parameters of *A. phloeocoptes*.

## Introduction

Apricot bud gall mite, *A. phloeocoptes* (Nalepa), is one of the most destructive pests found in apricot-producing regions worldwide, including central and southern Europe, the Mediterranean areas, North and South America, and East Asia [1]. This pest was first reported in 1890 by Nalepa [2] as plum pest. The occurrence of apricot bud gall mites in China was reported in 1991 [3]. Later, it was identified as *A. phloeocoptes* by Kuang, 1995 [4]. Apricot bud gall is

**Funding:** The National Key R&D Program of China (project 2016YFD0201100), and the Modern Fruit Industry Technology System of Gansu Province (GARS-SG-2). The funders had no role in study design, data collection and analysis, decision to publish, or preparation of the manuscript.

**Competing interests:** The authors have declared that no competing interests exist.

characterized by the malformation of apricot buds grown in various sizes which causes severe yield loss [5]. In recent years, the infestation by the apricot bud gall mites had become a significant threat to the development of the apricot industry in northwest China [6].

The fundamental challenge of pest management is to understand the life cycle of mite concerning time and space. What is the mite's life cycle and mechanisms of outbreak? The answers to these questions are not already available in the literature. *A. phloeocoptes* is a specie of Eriophyoidea, which is considered as the most diverse group of phytophagous mites [7], and are economically important pests [8]. They can disrupt the growth and physiology of host plants [9–11]. Population increase can be predicted by understanding mite's biological characteristics (egg, nymph, and adult).

Understanding the mode of dispersal is vital for managing populations of eriophyoid mites. Currently, only 2.5% of the approximately 4000 known eriophyoid mite transmission routes have been recorded [12, 13]. Eriophyoid mites have limited capacity in ambulatory movement because of the tiny body [14], the ability to disperse over long distances [15], or by airflow from the surface of the hosts [16]. What remains unclear about this field of research is the accurate understanding of ultrastructure of infested apricot bud gall, useful for assessing the degree of damage to invaded apricot trees.

It is suggested that phytohormones secreted by insects play a major role in gall formation and development [17]. The mechanism underlying the generation of insect gall was studied for a long time but remains largely unknown. The activity in various levels of the auxin indole-3-acetic acid (IAA) has been demonstrated in extracts of gall-forming homopterans [17]. The characteristic pattern of cell division and vascular-tissue development in galls of many insects argues that insect was the source of phytohormones in galls [18]. Recently it was suggested that auxin (IAA), cytokinin (CK), abscisic acid (ABA) and gibberellic acid (GA3) play a vital role in the induction and development of plant galls when infected by insects [19]. After reviewing the literature relevant to our research topic, we hypothesized that the phytohormones are mostly related to the development of galls in apricots and this needs scientific investigation.

The occurrence and spread of gall mite depend on abiotic factors and the resistance of the host plants [20]. Therefore, molecular methods are now being used to assess resistance. Investigation of different apricot cultivars for *A. phloeocoptes* infestation and identification of resistant cultivar and mechanism of resistance will help improve the disease management. This study aims to unravel some of the mysteries about the mode of dispersal of *A.phloeocoptes*, the development and ultrastructure of Apricot bud gall, and the role of phytohormones in the formation of apricot bud galls. All of our results were center on the interaction between apricot bud gall with their parasite. We tried to find reliable information which will be useful for designing future breeding programs for *A. phloeocoptes* resistance in Apricot.

## Materials and methods

### Field investigation and identification of apricot bud gall mite

Field evaluation of apricot bud gall mites was carried out on eight year old apricot orchards located in a popular apricot-producing area Qinwangchuan for three consecutive years from 2017 to 2019, Lanzhou City, China (36˚31' N, 103˚19 'E). The prevailing wind direction was northwest. Twenty apricot orchards in the area were selected for the study in a "Z" pattern. Five sites per apricot orchard were sampled, and each sample consisted of six arbitrarily selected trees. Therefore 30 apricot trees were sampled in each orchard. Ten branches were analyzed in each location (east, south, west, north, and center) per sampled tree. Each branch was observed according to the classification (S1 Table). The pest infestation index was

calculated based on the following formula:

$$Pest\ infestation\ index$$

$$= \frac{\sum_{k=0}^{4}(the\ number\ of\ branches \times quantitative\ value\ of\ the\ grade)}{the\ total\ number\ of\ branches \times the\ heaviest\ representative\ value} \times 100$$

Where $k$ is the $k$th quantitative value, and 4 is the number of all quantitative levels; the branch number represents the number of quantitative levels.

The morphological features of apricot bud gall mites were carefully observed using a Hitachi S-3400 scanning electron microscope and a stereomicroscope (SZX2-ILLTQ, TOKYO) in the laboratory. The abnormal buds were removed from the galls with tweezers and the surface of the abnormal buds were carefully cleaned with medical gauze dipped in alcohol. The distorted bud surface was dried with absolute alcohol. Young leaves were separated from the bud axis and were quickly pasted on the conductive tape. The sample was placed in the ion sputtering spraying instrument and sprayed with gold to make the sample conductive. The Samples were observed under electron microscope (Hitachi S-3400) [21].

Morphological components of different insect stages, including adults, nymphs, and eggs were studied using the electron microscope to determine the sample stages. Different parts of the mites were measured and photographed.

## Life cycle studies

To study the mite life cycle, overwintering locations, mite stages, and annual statistics of life stages were analyzed in several orchards in Qinwangchuan District during early March every year. Samples of gall, branches, bark cracks, and soil (up to 3-cm deep) around the mite-infested trees in the orchard were collected and kept into separate plastic bags for detailed observation under magnifying glass and stereo microscopes. Three galls were collected randomly from each branch, and the number of mites on ten young leaves per gall were recorded. The number of mites in different life stages was also calculated monthly.

To gain a better understanding of the mechanisms involved in *A. phloeocoptes* dispersal, We observed the passive dispersal modes included wind, animals and agricultural activities. Due to the tiny body, mites could be taken away by a strong wind from one tree to another or from one orchard to another. A transparent adhesive tape method was used. Sampled branches close to galls were wrapped with 1 cm wide transparent adhesive tape (the adhesive side outwards). Similarly, the trunk at 1 meter above ground was wrapped with 10 cm wide transparent adhesive tapes. Wrapped tapes were removed from the branches and brought to the laboratory for detailed investigation after 10 days. Sampling was performed once every 10 days, beginning from early March. Active transmission of eriophyes was determined by observing the sampled branches using an electron microscope (LEICA ICC50W). In early May, the potential spreading of eriophyes through insects, winds, rainfall, and farming practices were also recorded.

## Morphology of infested apricot bud

Morphological observation of infested apricot buds was carried out by paraffin sectioning [22], which is often used for anatomical analysis under an electron microscope (LEICA ICC50W). Infested apricot bud gall and healthy apricot buds of 2-year-old branches were used for this analysis. Healthy and infested buds were cut equally in two and immersed in Formalin Acetic acid Alcohol fixative solution (FAA) in a timely manner. Paraffin sectioning was performed by

following different steps including fixation, washing, dehydration, wax immersion, embedding, sectioning, sticking slice, wax dissolving, dyeing, hydrating, and sealing with gum [23]. The tissue structure of infested apricot bud gall and healthy apricot buds were compared using electron microscope (LEICA ICC50W) [24]. To support the ultrastructure details, we measured buds parameters, including the width of bud axis and the parameters of immature leaves.

Infested bud ultrastructure was observed under transmission electron microscopy (LSM 700) by ultrathin sectioning. Infested apricot bud gall and healthy apricot buds of 2-year-old branches were cut in half lengthwise and immersed in glutaraldehyde fixing solution. Ultrathin sectioning was performed through steps including fixation, rinsing, dehydration, saturation, embedding, resin polymerization, sectioning, and finally dying. Immature leaves and bud axes were observed [25, 26].

## Quantification of phytohormones

Infected and healthy plant materials were collected from naturally growing apricot trees in Qin-wangchuan District, China. Materials were collected from the field monthly during the growing season (on 30th of each month from april to september). Field collections were necessary because of the large numbers of galls needed for our analyses. Overall, more than 100 galls were taken to obtain sufficient tissues for extraction. All samples were frozen in liquid nitrogen and stored at -80°C until extraction. Samples (0.5 g dry weight each) were ground to a fine powder using mortars and extracted in 15 ml 80% methanol which was precooled at 4°C for 12 h. Then, centrifugation at 8000 rpm for 10 min at 4°C, remove the supernatant, add 5 ml precooled 80% methanol and incubated for 3 hours. Centrifugation at 8000 rpm for 10 min at 4°C. Decompress and concentrate the supernatant from the above two steps to remove methanol at 40°C, decolorize and extract in light petroleum ether for one time, discard petroleum phase, adjusted to pH 2.9, stored at 4°C. The whole experimental procedure was done under low light condition [27, 28]. Phytohormones including gibberellin (GA), zeatin (ZT), indoleacetic acid (IAA) and abscisic acid (ABA) in the tissue samples were quantified by HPLC (Agilent 1100) [29].

## Results

### Observed symptoms on apricot bud galls

Apricot bud gall mites mainly attacked the apricot buds, forming a typical bud gall. A gall consisted of numerous buds, which became brown with almost no sprouting or occasionally some weak buds sprouted and survived many years on the branch. The buds in the gall were found to be loose, hyperplastic, brittle, could be easily broken with tweezers, and the internal scales on the buds were not covered by external scales. Gall diameters were 1–4 cm on the 1-2-year-old apricot branches with dense bud growth. However, 5-8-cm galls occurred on severely damaged apricot branches with more than 20 buds (Fig 1).

It was observed that the flowering of infested apricot trees was delayed, the foliage became deformed, and fewer fruits of poor quality were produced, while occasionally, the whole tree exhibited stunted growth and, some of the trees died (Fig 1). We observed that apricot bud gall occurred more severely in densely planted orchards than those with wide spacing. Intensive orchard management reduced the chances of disease onset. It was observed that Apricot bud galls were more prevalent on 2-3-year-old branches than older branches.

### Morphological features of apricot bud gall mites

Electron microscopic studies identified apricot bud gall mites as *A. phloeocoptes*. The adult female bodies were vermiform and ivory, 177.38–265.67 μm long, and 44.11–60.59 μm wide.

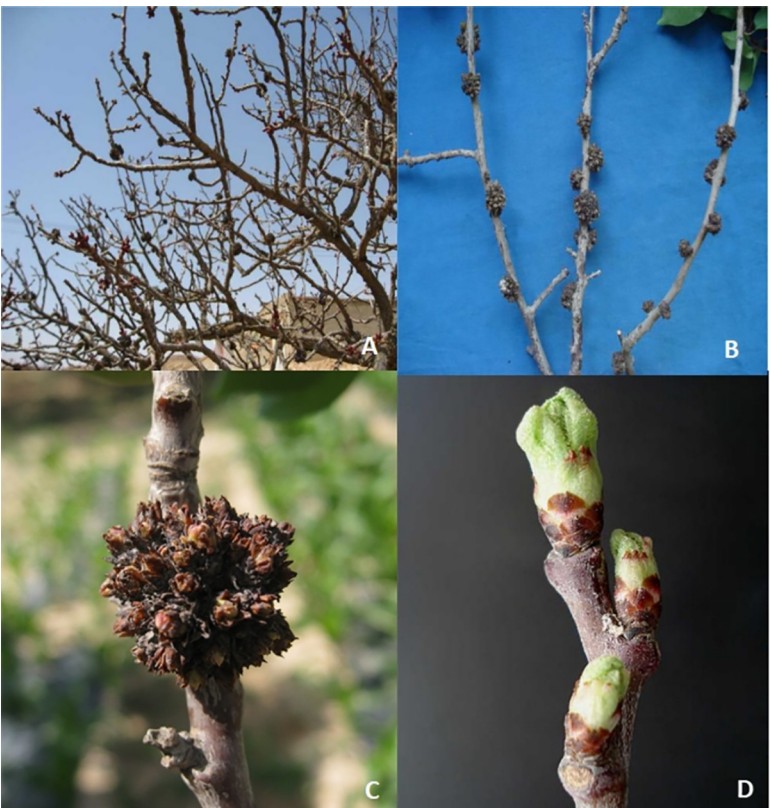

**Fig 1. Apricot bud gall symptoms.** Apricot tree infested by *A. phloeocoptes* (Nalepa), (A) different size of galls formed in branches. (B) Galls on one-two year old apricot branches, the galls diameter were1-4 cm. (C) An enlarged apricot bud gall on a severely damaged apricot branch formed by more than 20 buds. (D) A branch with healthy buds.

The anterior part of the body was slightly wider and carrot-shaped (Fig 2A). The female genital is 16.780.59 μm wide. The ant 90–26.85 μm, wide and the genital cover had flap distributed grain (Fig 2B). The feather claw had five lateral ramifications (Fig 2D), with two pairs of legs

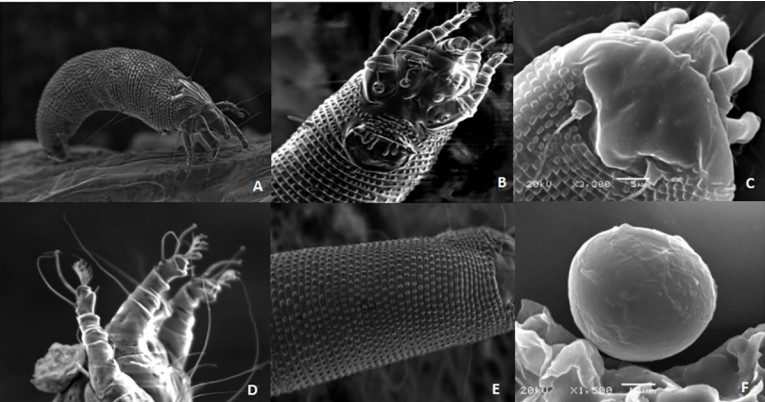

**Fig 2. Body view of an adult female and egg.** (A) Female adult of *A. phloeocoptes* (Nalepa), The body is *vermiform and milky* white in color. The adult female is 48.26 μm long and 33.63 μm wide. (B) Head of the mite, Below the foot base is external genitalia. There are grain dots on genital cover flap. (C) Isosceles triangle shaped dorsal shield of the mite, smooth and without anterior lobe, and has two pairs of legs. (D) Feather claw of the mite. The claws are feathery and have five lateral branches. (E) Lateral microtubercles of the mite. (F) Egg are oval shaped and white in color.

Table 1. Morphological characters of *Acalitus phloeocoptes*.

| Morphological Features | Measurements |
|---|---|
| Adult female body | 177.38–265.67 µm long, 44.11–60.59 µm wide |
| Female genital | 16.78µm -17.44µm long and 24.90–26.85µm wide |
| Dorsal shield | 20–25.6 µm long and 34.68–40.60 µm wide |
| Dorsal seta | 20 µm long |
| Eggs | 48.26–53.31µm long and 33.63–40.12 µm wide |

and three pairs of coxal setae. The dorsal shield resembles an isosceles triangle, smooth and without anterior lobe, which is 20–25.6 µm long and 34.68–40.60 µm wide. The dorsal tubercle was located at the back edge of the dorsal shield (Fig 2C), while the body section was divided into three parts. The dorsal seta was found to be 20 µm long and projected backward diagonally. Tergites of the body is arched with 64 lateral microtubercles. The adult female is 48.26 µm long and 33.63 µm wide (Fig 2E). The eggs were white and oval-shaped, approximately 48.26–53.31µm long and 33.63–40.12 µm wide (Fig 2F). Male adults were not collected in the present experiment because they were not found (Table 1).

Ultrathin sectioning of infested apricot buds under electron microscopy revealed an additional ultrastructure detail. The anatomical structures of the attacked apricot tree buds were different from those of the healthy buds (Fig 3D) and whose axis had only one bud, but an infested bud was deformed and had more than one bud. The bud primordium was covered by 1–3 layers of lamellar structure during the initial stages of apricot bud gall formation (Fig 3A). At the mid-stage, the bud primordium grew into a new bud while (Fig 3B), at a later stage, a bud axis branched out to form approximately 4–5 bud axes, and each axis was covered by the layer of young leaves, an apricot bud gall was formed afterward (Fig 3C). Careful measurements of infested buds demonstrated that the width of the lower part of the bud axis was 1500.8 µm whereas the healthy ones were 332.0 µm. Infested buds had more immature leaves than healthy buds, infested buds had 24 leaves however, healthy buds had 12 leaves (S2 Table).

## The regularity outbreak of *A. Phloeocoptes* and their disperse mode

The study of the life cycle revealed that *A. phloeocoptes* mainly survived through winter in tight, live galls. Adults accounted for 70.4% of all mites, whereas nymphs and eggs accounted for 15.1% and 12.6%, respectively (S3 Table); Mites were mostly found in galls. In branches and cracked

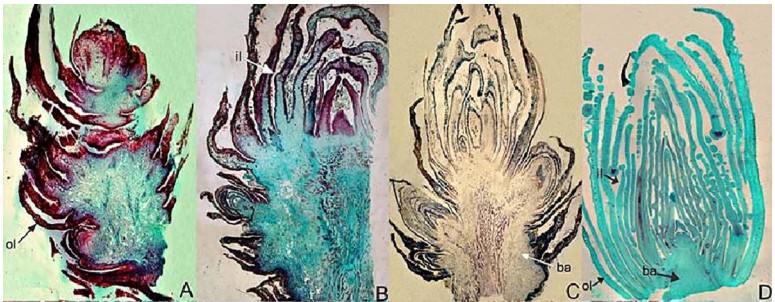

**Fig 3. Deformation process of an apricot acanthoid gall.** (A) Initial stage of apricot bud gall and bud primordium. The bud primordium increased significantly at the axil of young leaves. (B) Mid-stage of apricot bud gall and bud primordium. (C) Late stage of apricot bud gall and bud primordium. Multiple bud cluster structure were formed. (D) healthy apricot bud. No lignification was observed in the whole bud. Inner leaf, outer leaf, and bud axis were represented by il, ol, and ba respectively.

**Table 2. Annual statistic of life stages.**

| Life stages \\ Month | 1 | 2 | 3 | 4 | 5 | 6 | 7 | 8 | 9 | 10 | 11 | 12 |
|---|---|---|---|---|---|---|---|---|---|---|---|---|
| Egg | 4.92 | 5.54 | 5.16 | 5.09 | 11.92 | 7.15 | 6.03 | 6.85 | 5.40 | 6.32 | 6.31 | 5.35 |
| Nymph | 11.36 | 12.04 | 14.36 | 14.30 | 18.80 | 25.70 | 16.67 | 18.66 | 18.46 | 17.73 | 15.82 | 13.33 |
| Adult | 37.00 | 39.74 | 33.76 | 38.02 | 42.85 | 46.18 | 63.06 | 44.77 | 48.73 | 41.19 | 35.63 | 43.74 |

bark, 0.6% of only adults were found. mites were not found in the surrounding soil. Therefore, these mites mainly survive over winter in tight, live galls as adults and a few eggs (S3 Table).

The number of mites at each life stage (egg, nymph, and adult) was counted under the stereo microscope. Oviposition peaked in May, with the highest value of 10.864 eggs (Table 2); the number of nymphs was highest in June, with a maximum value of 27.361 nymphs; and the number of adults peaked in July, with the highest value of 61.362 adults. The number of adults decreased with the decrease in temperature during September (Table 2).

Our results indicated that wind was one of the passive dispersal modes of the mite. In addition, animal carries were another important dispersal mode. Results also showed that the mites could attach themselves to other animals, such as ostracum of *Pseudaulacaspis pentagona* (Targioni Tozzetti) and *Didesmococcus koreanus Borchsenius*, which can be found around the apricot bud galls. A large number of scale insect nymphs could be found in apricot bud galls (Fig 4), and many adults, nymphs, and eggs of *A. phloeocoptes* attached to the body of insects were found as well. Moreover, there were many nymphs of *Chilocorus rubidus* hosted on the back of apricot leaves, many eggs of *A. phloeocoptes* accreted with the body of these nymphs.

## Ultra-structural changes of infested apricot bud

**Ultrastructure of infested bud.** Examination of the affected buds under transmission electron microscopy showed that during the early stage of infestation, starch granules in bud axon were expanded and occupied a significant portion of chloroplast space as compared with those found in healthy buds (Fig 5A and 5B). The cytoplasm seemed to be swollen and deformed. Many small vacuoles in the cell were distributed on their edges (Fig 5C). In the later stages of infestation, the cytoplasm of each infected cell was wholly disrupted, with severely deformed chloroplasts, mitochondria and irregular vacuoles (Fig 5D).

**Ultrastructure of affected bud leaf.** The cell wall of the apricot bud leaf in infested cells was thicker (S1B Fig) than the healthy cells (S1A Fig). The cytoplasm was denser and deeper in color, mitochondrion increased in number and became slightly larger than those in the unaffected ones (S1C Fig). In the early stage of injury, starch granules expanded and extruded from the granal lamellae to the edge of chloroplast. The inner side of cell wall was deposited with a thin, electron-dense substance that was closely connected to the inside of the cell wall, and the vacuole was muddy and dispersed with a flocculent substance. Finally, the organelles disintegrated and many infected cells collapsed and were filled with a compacted layer of amorphous, moderately electron-opaque material (S1 Fig).

**Role of endogenous hormones in gall formation.** To better understand *A. phloeocoptes* infestation, various hormone contents (ZT, IAA, ABA, and GA3) were quantified at different stages of infestation. Maximum ZT content was observed during May and June in healthy and gall buds respectively (Fig 6A).

Similarly, levels of IAA had the same trend as ZT, maximum IAA was observed during June and May in healthy and gall buds respectively (Fig 6B). We found that IAA level from gallbuds was far lower than that from healthy buds. The levels of ABA from gall buds were significantly higher than healthy buds during the period of study (Fig 6C). Levels of GA3 were

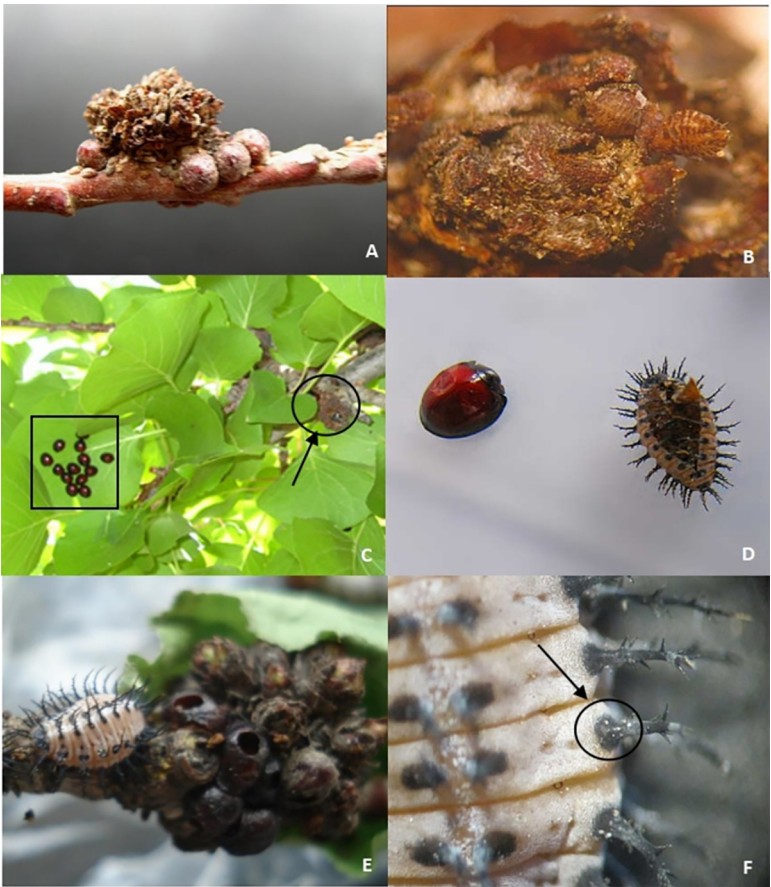

**Fig 4. Passive spread of *A. phloeocoptes*.** (A) The apricot bud gall is surrounded by a necklace like scale insect shell. (B) Nymphs of *Pseudaulacaspis pentagona* in the scale layers of apricot bud gall. (C) *Chilocorus rubidus* adults lived on the back of apricot leaves. (D) Adult and nymph of *C. rubidus*. (E) Nymph of *C.rubidus close to* apricot bud gall. (F) *A. phloeocoptes* eggs on *C.rubidus* nymph.

also affected in gall buds compared to healthy buds during the growing season (Fig 6D). The GA3 content of infested buds had a sharp increase (the increment was 154.6%) from april to may and reached maximum levels in may, whereas GA3 in healthy buds demonstrated a slow increase (the increment was 37.2%) at the same time. Levels of GA3 in infested buds were much less than those in healthy buds in September.

## Discussion

Apricot bud gall infestation by *A. phloeocoptes* is complex, not only because of the causal organism but also because the pests are localized in buds and have special dispersal mode. Eriophyoid mites can have a serious impact on plant health, it was previously showed that *Aceria sheldoni chinensis* cause damage to Citrus sprouts [30] and *Acalitus persicae* can infest peach bud and cherry bud [31]. These gall mites cause damage to bud scales, the injured buds expanded to form bud galls, and could not germinate, spread leaves and blossom normally, Finally, the buds died. The eriophyoid mite (*A. phloeocoptes*) had serious impact on the apricot orchard health since it can lead severe deformity of buds and eventually leads to the death of the plant. What remains unclear about this field of research is the life cycle of *A. phloeocoptes*, the regularity in its outbreak, dispersal, overwintering locations, mite stages, and annual

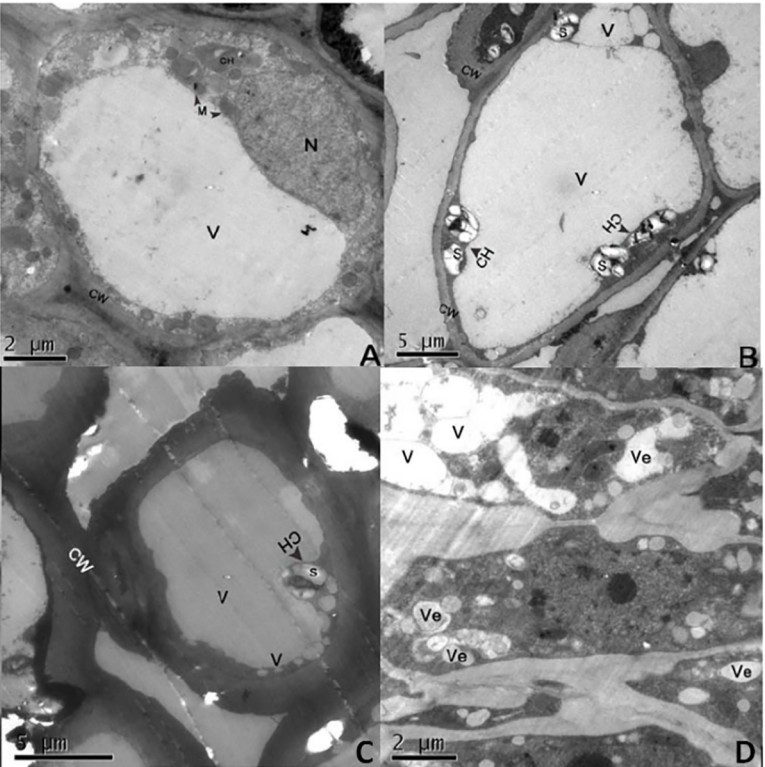

**Fig 5. Ultrastructure of apricot bud gall bud axis cells.** (A) Healthy cell. (B) Early stage of injury period: the starch granules in the axon of the injured bud were obviously expanded, there is a large liquid bubble and a few small vacuoles in the cell, the cell fluid is clear. (C) Mid stage of injury: Many vacuoles in the cell are distributed on the edge of the cell. Some vacuoles contain cytoplasmic content, and the cytoplasm seems to be engulfed by vacuoles. (D) Late stage of injury: The nucleus, chloroplast and mitochondrion were seriously deformed, protoplasm became concentrated, and there was no obvious organelle in the cell, but filled with round or irregular vacuoles. CH: chloroplast; CY: cytoplasm; CW: cell wall; M: mitochondrion; N: nucleus; S: starch; V: vacuole; Ve: vesicle; *:electron-dense granules.

statistics of life stages, and all these were analyzed in several orchards in our current study. Our results suggested that these mites mainly survive over winter in tight, live galls as adults with few eggs. Furthermore, we have revealed the passive dispersal modes of *A. phloeocoptes* that were wind, animals and agricultural practices. One of the worth mentioning modes is the animal, which would be far more efficient than any other dispersal modes. Karasawa and Lindo reported a major factor of the colonization of arboreal habitats by oribatid mites which was wind dispersal [32, 33] and was partially consistent with our results.

Little was known about the effect of mites on the ultrastructure of the host. We have focused on the effects of mites on the ultrastructure of host tissues. We have found that cytoplasm of each infected cell was completely disrupted, with severely deformed chloroplasts, mitochondria and irregular vacuoles of an infested bud during later stages of infestation by *A. phloeocoptes*. Previously, LanJinghua [34] reported that *Tetranychus cinnabar* caused partial or complete disappearance of chloroplasts from cotton leaves, disordered arrangement of grana lamellae and increased starch grains. Starch grains are chloroplast photosynthates. When cells are damaged, starch grains cannot be transported to the storage sites normally, which increases their number and accumulation. The starch grains in cotton leaf cells increased significantly after *Tetranychus cinnabar* infestation [35]. Vacuoles are the largest organelles in plant cells, which are filled with cell fluid to maintain cell morphology and store ions and metabolites.

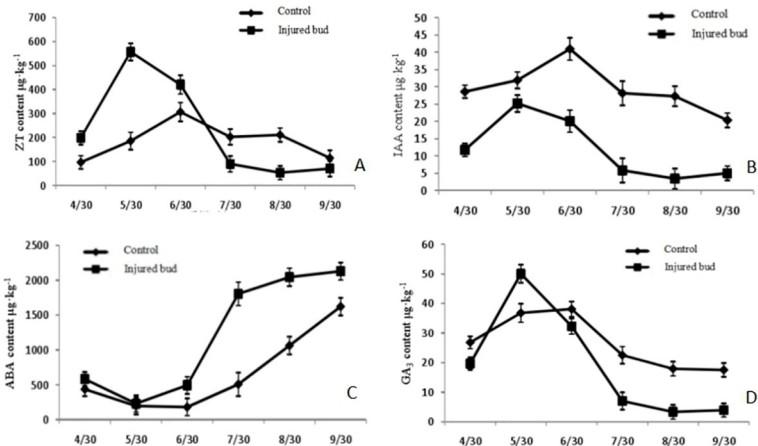

**Fig 6. Effects of *A. phloeocoptes* on phytohormones content in apricot bud gall.** (A) ZT contents quantified during May and October in healthy and gall buds. (B) IAA content quantified during June and May in healthy and gall buds. (C) The ABA content of healthy and gall buds. ABA content was significantly higher than healthy buds during the period of study. (D) The GA content of healthy and gall bud. Infested buds showed a sharp increase of GA content from April to May and reached maximum levels in May, while GA in healthy buds demonstrated a slow increase at the same time.

Vacuoles can engulf useless organelles. When cells are damaged, vacuoles rupture and release black osmiophilic substances, which accumulate in cells. In addition, rapid nuclear degradation is triggered by vacuole rupture [36].

Phytohormones have been speculated to play a vital role in symptom development, yet their precise role was not known. In the previous decade, a large number of studies were carried out to understand the role of phytohormones in plant growth, development, and tolerance against various biotic and abiotic stress [37]. however, it was reported that phytohormones play a key role in the successful infection by affecting insect behavior. Nematode-induced feeding sites show dramatic changes in host cell morphology and gene expression. These changes are likely mediated, at least in part, by phytohormones [38]. Our study has demonstrated that phytohormones played a vital role in the mechanism of gall formation. Accumulation of large amounts of ZT and IAA leads to rapid bud proliferation during rapid growth period. It is well known that ZT promotes cell division [39]. Therefore, the accumulation of large amounts of ZT led to the rapid bud proliferation during rapid growth period, indicating that within the period from May 30th to June 30th, there was increased vegetative growth in gall buds, which is the initial stage of gall formation.

ABA controls bud development and results in quick maturation [40]. The phytohormone ABA regulates developmental processes and also has a primary function in accelerating aging [41]. Accordingly, the high content of ABA affects the development of apricot buds. Lower levels of GA3 leads to quick lignification at later phases of development [42]. Infection results in 2–4 times less accumulation of IAA and 2.5–5 times more accumulation of GA3 in infected tissue compared to healthy tissues. GA3 can promote cell elongation and cell division [43]; therefore, gall buds with bigger diameter were formed by the severe distortion and enlargement inside the bud cells. Levels of GA3 in infested buds were much less than those in healthy buds in September, indicating that the lower content of GA3 led to rapid lignification at later stages of the growth. In conclusion, the difference in hormone content suggests their role in infection and symptom development [44]. Overall, our results have revealed that the mechanism underlying the interaction of apricot bud gall with its parasite and provide reliable information for

designing valuable Apricot breeding programs. This study will be quite useful for pest management and will provide a comprehensive evaluation of ecology-based cost-effective control, life history and demographic parameters of *A. phloeocoptes*.

## Supporting information

**S1 Fig. Ultrastructure of apricot bud gall immature leaf cells.** (A) Early stage of injury: the cell wall of young leaves was obviously thickened, a small amount of starch granules were formed, and the cell fluid in vacuole was clear and bright. Mid stage of injury: (B) Mitochondria are obviously expanded and irregular in shape (C) on the inner side of the cell wall there is a thin layer of electron dense material deposition, which is closely connected with the inner side of the cell wall. (D) Late stage of injury: The organelles in the cells disintegrated, and the substances with high electron density scattered in the cells. **CH**: chloroplast; **CY**: cytoplasm; **CW**: cell wall; **M**: mitochondrion; **N**: nucleus; **S**: starch; **V**: vacuole. **Ve**: vesicle; *:electron-dense granules.
(JPG)

**S1 Data.**
(XLSX)

**S1 Table. Grading standard of apricot bud galls.**
(DOCX)

**S2 Table. Differences between infested bud and healthy bud.**
(DOCX)

**S3 Table. Number of acalitus phloeocoptes (Nalepa) in different stages.** Measured from various overwintering places in in three consecutive years in Lanzhou City, Gansu Province, China.
(DOCX)

## Acknowledgments

We thank Professor Meiliang Zhou for critical reviews of this manuscript. We thank Jerry for advice in the statistical analyses and his critical reviews about the manuscript.

## Author Contributions

**Data curation:** Jin Zhang.

**Formal analysis:** Zeshan Hassan, Shahbaz Ahmad.

**Investigation:** Mohammed Mujitaba Dawuda.

**Methodology:** Junsheng Yao.

**Supervision:** Muhammad Khurshid, Bingliang Xu.

**Writing – original draft:** Shijuan Li.

**Writing – review & editing:** Shijuan Li.

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
