## [Decision Letter · Decision Letter 0]

9 Jun 2021

PONE-D-21-11552

Interaction of the Causal Agent of Apricot bud gall Acalitus phloeocoptes (Nalepa) with Apricot: Implications in Infested Tissues

PLOS ONE

Dear Dr. Shijuan,

Thank you for submitting your manuscript to PLOS ONE. After careful consideration, we feel that it has merit but does not fully meet PLOS ONE’s publication criteria as it currently stands. Therefore, we invite you to submit a revised version of the manuscript that addresses the points raised during the review process.

We look forward to receiving your revised manuscript.

Kind regards,

Xiao-Yue Hong, Ph.D

Academic Editor

PLOS ONE

Journal Requirements:

2) In your Methods section, please provide additional location information of the study site, including geographic coordinates for the data set if available.

3) In your Methods section, please provide additional information regarding the permits you obtained for the work. Please ensure you have included the full name of the authority that approved the study site access and, if no permits were required, a brief statement explaining why.

4)  Thank you for stating the following in the Acknowledgments Section of your manuscript:

[This research was supported in part by the National Key R＆D Program of China (project 2016YFD0201100),

and by the Modern Fruit Industry Technology System of Gansu Province (GARS-SG-2). We thank Professor

Meiliang Zhou for advice and his critical reviews of this manuscript; thank Professor Panos for his critical editing

suggestions of the English language; thank Jerry for advice in the statistical analyses and his critical reviews about

manuscript. This study was supported by the College of Plant Protection, Gansu Agriculture University, China.]

 [National Key R＆D Program of China (project 2016YFD0201100), and by the Modern

Fruit Industry Technology System of Gansu Province (GARS-SG-2)]

Reviewers' comments:

Reviewer's Responses to Questions

**Comments to the Author**

1. Is the manuscript technically sound, and do the data support the conclusions?

Reviewer #1: Yes

2. Has the statistical analysis been performed appropriately and rigorously? 

Reviewer #1: Yes

3. Have the authors made all data underlying the findings in their manuscript fully available?

Reviewer #1: Yes

4. Is the manuscript presented in an intelligible fashion and written in standard English?

Reviewer #1: No

5. Review Comments to the Author

Reviewer #1: This is a thorough analysis of Apricot bud gall mites with their hosts. The major contribution of the authors is to expand upon our understanding of interactions of apricot bud gall with its parasite and provide reliable information for breeding. However, the manuscript is written in very simple English and some sentences should be rewritten or reorganized. Some parts of text are mixed and should be moved to other chapters.

1. For Introduction the third part, “Lanjinghua，reported the ultrastructural changes in cotton leaves affected by Carmine spider mite and showed that the disintegrated organelles and incomplete chloroplast membrane”. Please check the grammar and citation of “Lanjinghua”. Further, no reports for spider mites infest host to form galls, which should be different to the gall-forming of eriophyoid mites. Delete this sentence and the following sentences.

2. For Material and Methods. Provide the procedure of electron microscope (Hitachi S-3400) in detail. Provide the full name of “HPLC”. Provide the date of field surveys.

3. For the description of morphological characters of Acalitus phloeocoptes, the authors should follow the terminology of Amrine et al. (2003) or Lindquist (1996). A table should be included for all the measurements if they are different from previous report.

4. “We observed that the passive dispersal modes of A. phloeocoptes include wind, animals and agricultural activities. Due to the tiny body, mites could be taken away by a strong wind from one tree to another or from one orchard to another. When observed under the microscope in the laboratory, the transparent adhesive tape wrapped on the trunk 1 meter above the ground captured more mites on the windward side, which indicated that wind was one of the passive dispersal modes of the mite.” This part should be moved to the Methods.

5. “These kinds of eriophyoid mites have a serious impact on the apricot orchard health since it leads a severe deformity of buds and eventually leads to the death of the plant.” For A. phloeocoptes or other eriophyoid mites? Name it and their references.

6. For Discussion paragraph two, “Ultrastructural studies of pathogens were earlier applied to observe the structure and infection process of rust fungi.” There is no relationship of key findings with fungi.

6. PLOS authors have the option to publish the peer review history of their article (what does this mean?). If published, this will include your full peer review and any attached files.

Reviewer #1: No

---

## [Author Response · Author response to Decision Letter 0]

12 Jul 2021

Response to Reviews

Journal Requirements:

Thank you for your valuable advice, we revised the format of the manuscript to meet PLOS ONE's style requirements. 

2) In your Methods section, please provide additional location information of the study site, including geographic coordinates for the data set if available.

We added geographic coordinates in methods section marked in red. (line 84-85). 

3) In your Methods section, please provide additional information regarding the permits you obtained for the work. Please ensure you have included the full name of the authority that approved the study site access and, if no permits were required, a brief statement explaining why. 

No permits were required from the site access, the investigation site was the research base of Gansu Agriculture University. All of the work were carried out with verbal permission. An unknown disease occurred in apricot orchards in the research base. SJ, Li and J, Zhang did the preliminary investigation and identification of the pathogen. 

4) Thank you for stating the following in the Acknowledgments Section of your manuscript:

[This research was supported in part by the National Key R＆D Program of China (project 2016YFD0201100),

and by the Modern Fruit Industry Technology System of Gansu Province (GARS-SG-2). We thank Professor Meiliang Zhou for advice and his critical reviews of this manuscript; thank Professor Panos for his critical editing suggestions of the English language; thank Jerry for advice in the statistical analyses and his critical reviews about manuscript. This study was supported by the College of Plant Protection, Gansu Agriculture University, China.]

 [National Key R＆D Program of China (project 2016YFD0201100), and by the Modern

Fruit Industry Technology System of Gansu Province (GARS-SG-2)] 

Reviewers' comments:

Reviewer's Responses to Questions

We removed funding-related text from the manuscript. Funding information can present in the Funding Statement section of the online submission form. The Funding Statements were: [National Key R＆D Program of China (project 2016YFD0201100), and by the Modern Fruit Industry Technology System of Gansu Province (GARS-SG-2)], which were added in cover letter. 

5. Review Comments to the Author

Reviewer #1: This is a thorough analysis of Apricot bud gall mites with their hosts. The major contribution of the authors is to expand upon our understanding of interactions of apricot bud gall with its parasite and provide reliable information for breeding. However, the manuscript is written in very simple English and some sentences should be rewritten or reorganized. Some parts of text are mixed and should be moved to other chapters. 

1. For Introduction the third part, “Lanjinghua，reported the ultrastructural changes in cotton leaves affected by Carmine spider mite and showed that the disintegrated organelles and incomplete chloroplast membrane”. Please check the grammar and citation of “Lanjinghua”. Further, no reports for spider mites infest host to form galls, which should be different to the gall-forming of eriophyoid mites. Delete this sentence and the following sentences. 

Based on the advice you gave, we deleted this sentence and the following sentences.

2. For Material and Methods. Provide the procedure of electron microscope (Hitachi S-3400) in detail. Provide the full name of “HPLC”. Provide the date of field surveys.

We added the procedure in material and methods section of manuscript marked in red (line 95-102 ). 

Full name of “HPLC” was marked in red (line 156).

The date of field surveys: the surveys were conducted for three consecutive years from January 2017 to December 2019. 

3. For the description of morphological characters of Acalitus phloeocoptes, the authors should follow the terminology of Amrine et al. (2003) or Lindquist (1996). A table should be included for all the measurements if they are different from previous report. 

We added a Table 1 for all the measurements.

4. “We observed that the passive dispersal modes of A. phloeocoptes include wind, animals and agricultural activities. Due to the tiny body, mites could be taken away by a strong wind from one tree to another or from one orchard to another. When observed under the microscope in the laboratory, the transparent adhesive tape wrapped on the trunk 1 meter above the ground captured more mites on the windward side, which indicated that wind was one of the passive dispersal modes of the mite.” This part should be moved to the Methods.

We moved this part to the Methods. In addition, we modified the sentences in the methods and result part, which were marked in red (line 114-117 and line 213-216). 

5. “These kinds of eriophyoid mites have a serious impact on the apricot orchard health since it leads a severe deformity of buds and eventually leads to the death of the plant.” For A. phloeocoptes or other eriophyoid mites? Name it and their references.

We added references and other eriophyoid mites in MS marked in red. (line 254-262 ). 

6. For Discussion paragraph two, “Ultrastructural studies of pathogens were earlier applied to observe the structure and infection process of rust fungi.” There is no relationship of key findings with fungi.

We deleted these sentences which are not related to our findings. 

6. PLOS authors have the option to publish the peer review history of their article (what does this mean?). If published, this will include your full peer review and any attached files.

Do you want your identity to be public for this peer review? For information about this choice, including consent withdrawal, please see our Privacy Policy. 

Reviewer #1: No 

---

## [Editor Report · Decision Letter 1]

12 Aug 2021

Interaction of the Causal Agent of Apricot bud gall Acalitus phloeocoptes (Nalepa) with Apricot: Implications in Infested Tissues

PONE-D-21-11552R1

Dear Dr. Shijuan,

We’re pleased to inform you that your manuscript has been judged scientifically suitable for publication and will be formally accepted for publication once it meets all outstanding technical requirements.

Kind regards,

Xiao-Yue Hong, Ph.D

Academic Editor

PLOS ONE
---

## [Editor Report · Acceptance letter]

23 Aug 2021

PONE-D-21-11552R1 

Interaction of the causal agent of Apricot bud gall *Acalitus phloeocoptes* (Nalepa) with apricot: Implications in infested tissues 

Dear Dr. Li:

I'm pleased to inform you that your manuscript has been deemed suitable for publication in PLOS ONE. Congratulations! Your manuscript is now with our production department. 

Kind regards, 

on behalf of

Dr. Xiao-Yue Hong 

Academic Editor

PLOS ONE